# Communication Efficient Federated Learning for Generalized Linear Bandits

**Chuanhao Li**[1]    **Hongning Wang**[1]
[1]Department of Computer Science, University of Virginia
`{cl5ev,hw5x}@virginia.edu`

## Abstract

Contextual bandit algorithms have been recently studied under the federated learning setting to satisfy the demand of keeping data decentralized and pushing the learning of bandit models to the client side. But limited by the required communication efficiency, existing solutions are restricted to linear models to exploit their closed-form solutions for parameter estimation. Such a restricted model choice greatly hampers these algorithms' practical utility. In this paper, we take the first step to addressing this challenge by studying generalized linear bandit models under the federated learning setting. We propose a communication-efficient solution framework that employs online regression for local update and offline regression for global update. We rigorously proved, though the setting is more general and challenging, our algorithm can attain sub-linear rate in both regret and communication cost, which is also validated by our extensive empirical evaluations.

## 1 Introduction

As a classic model for sequential decision making problems, contextual bandit has been widely used for a variety of real-world applications, including recommender systems [19], display advertisement [21] and clinical trials [7]. While most existing bandit solutions are designed under a centralized setting (i.e., data is readily available at a central server), in response to the increasing application scale and public concerns of privacy, there is increasing research effort on federated bandit learning lately [28, 6, 27, 11, 17], where $N$ clients collaborate with limited communication bandwidth to minimize the overall cumulative regret incurred over a finite time horizon $T$, while keeping each client's raw data local. Compared with standard federated learning [23, 13] that works with fixed datasets, federated bandit learning is characterized by its online interactions with the environment, which continuously provides new data samples to the clients over time. This brings in new challenges in addressing the conflict between the need of timely data/model aggregation for regret minimization and the need of communication efficiency with decentralized data. A carefully designed model update method and communication strategy become vital to strike this balance.

Existing federated bandit learning solutions only partially addressed this challenge by considering simple bandit models, like context-free bandit [27] and contextual linear bandit [28, 6, 17], where closed-form solution for both local and global model update exists. Therefore, efficient communication for global bandit model update is realized by directly aggregating local sufficient statistics, such that the only concern left is how to control the communication frequency over time horizon $T$. However, such a solution framework does not apply to the more complicated bandit models that are often preferred in practice, such as generalized linear bandit (GLB) [8] or neural bandit [30], where only iterative solutions exist for parameter estimation (e.g., gradient-based optimization). To enable joint model estimation, now the learning system needs to solve distributed optimization for multiple times as new data is collected from the environment, and each requires iterative gradient/model aggregation among clients. This is much more expensive compared with linear models, and it naturally leads to the question: whether a communication efficient solution to this challenging problem is still possible?

36th Conference on Neural Information Processing Systems (NeurIPS 2022).

In this paper, we answer this question affirmatively by proposing the first provably communication efficient algorithm for federated GLB that only requires $\tilde{O}(\sqrt{T})$ communication cost, while still attaining the optimal order of regret. Our proposed algorithm employs a combination of online and offline regression, with online regression adjusting each client's model using its newly collected data, and offline (distributed) regression occasionally soliciting local gradients from all $N$ clients for joint model estimation when sufficient amount of new data has been accumulated. In order to balance exploration and exploitation in arm selection, we propose a novel way to construct the confidence set based on the sequence of *offline-and-online* model updates that each client has received. The initialization of online regression with offline regression introduces dependencies that break the standard martingale argument, which requires proof techniques unique to this paper.

We also explored other non-trivial solution ideas to further justify our current design. Specifically, in practice, a common way to update the deployed model for applications with streaming data is to set a schedule and periodically re-train the model using iterative optimization methods. For comparison, we propose and rigorously analyze a federated GLB algorithm designed based on this idea, as well as a variant that further enables online updates on the clients. We also consider another solution idea motivated by distributed/batched online convex optimization, which is characterized by lazy online updates over batches of data. Moreover, extensive empirical evaluations on both synthetic and real-world datasets are performed to validate the effectiveness of our algorithm.

## 2 Related Work

GLB, as an important extension of linear bandit models, has demonstrated encouraging performance in modeling binary rewards (such as clicks) that are ubiquitous in real-world applications [18]. The study of GLB under a centralized setting dates back to Filippi et al. [8], who proposed a UCB-type algorithm that achieved $\tilde{O}(d\sqrt{T})$ regret. Li et al. [20] later proposed two improvements: a similar UCB-type algorithm that improves the result of [8] by a factor of $O(\log T)$, which has been popularly used in practice as it avoids the projection step needed in [8]; and another impractical algorithm that further improves the result by a factor of $O(\sqrt{d})$ assuming fixed number of arms. To improve the time and space complexity of the aforementioned GLB algorithms, followup works adopted online regression methods. In particular, motivated by the online-to-confidence-set conversion technique from [2], Jun et al. [12] proposed both UCB and Thompsan sampling algorithms with online Newton step, and Ding et al. [4] proposed a Thompson sampling algorithm with online gradient descent, which, however, requires an additional context regularity assumption to obtain a sub-linear regret.

GLB under federated/distributed setting still remains under-explored. The most related works are the federated/distributed linear bandits [16, 28, 6, 11, 17]. In these works, thanks to the existence of closed-form solution for linear models, the clients only communicate their local sufficient statistics for global model update. Korda et al. [16] considered a peer-to-peer (P2P) communication network and assumed the clients form clusters, i.e., each cluster is associated with a unique bandit problem. But as they only focused on reducing *per-round* communication, the communication cost is still linear over time. Huang et al. [11] considered a star-shaped communication network as in our paper, but their proposed phase-based elimination algorithm only works in fixed arm set setting. The closest works to ours are [28, 6, 17], which uses event-triggered communication protocols to obtain sub-linear communication cost over time for federated linear bandit with a time-varying arm set.

Another related line of research is the standard federated learning that considers offline supervised learning problems [13]. Since its debut in [23], FedAvg has become the most popularly used algorithm for offline federated learning. However, despite its popularity, several works [22, 14, 24] identified that FedAvg suffers from a *client-drift* problem when the clients' data are non-IID (which is an important signature of our case), i.e., local iterates in each client drift towards their local minimum. This leads to a sub-optimal convergence rate of FedAvg: for example, one has to suffer a sub-linear convergence rate for strongly convex and smooth losses, though a linear convergence rate is expected under a centralized setting. To alleviate this, Pathak and Wainwright [26] proposed an operator splitting procedure to guarantee linear convergence to a neighborhood of the global minimum. Later, Mitra et al. [24] introduced variance reduction techniques to guarantee exact linear convergence to the global minimum.

# 3 Preliminaries

In this section, we first introduce the general problem formulation of federated bandit learning, and discuss the existing solutions under the linear reward assumption. Then we formulate the federated GLB problem considered in this paper, followed by detailed discussions about the new challenges compared with its linear counterpart.

## 3.1 Federated Bandit Learning

Consider a learning system with 1) $N$ clients responsible for taking actions and receiving corresponding reward feedback from the environment, e.g., each client being an edge device directly interacting with a user, and 2) a central server responsible for coordinating the communication between the clients for joint model estimation.

At each time step $t = 1, 2, ..., T$, all $N$ clients interact with the environment in a round-robin manner, i.e., each client $i \in [N]$ chooses an arm $\mathbf{x}_{t,i}$ from its time-varying candidate set $\mathcal{A}_{t,i} = \{\mathbf{x}_{t,i}^{(1)}, \mathbf{x}_{t,i}^{(2)}, \ldots, \mathbf{x}_{t,i}^{(K)}\}$, where $\mathbf{x}_{t,i}^{(a)} \in \mathbb{R}^d$ denotes the context vector associated with the $a$-th arm for client $i$ at time $t$. Without loss of generality, we assume $||\mathbf{x}_{t,i}^{(a)}||_2 \leq 1, \forall i, a, t$. Then client $i$ receives the corresponding reward $y_{t,i} \in \mathbb{R}$ from the environment, which is drawn from the reward distribution governed by an unknown parameter $\theta_\star \in \mathbb{R}^d$ (assume $\|\theta_\star\| \leq S$), i.e., $y_{t,i} \sim p_{\theta_\star}(y|\mathbf{x}_{t,i}^{(a)})$. The interaction between the learning system and the environment repeats itself, and the goal of the learning system is to minimize the cumulative (pseudo) regret over all $N$ clients in the finite time horizon $T$, i.e., $R_T = \sum_{t=1}^T \sum_{i=1}^N r_{t,i}$, where $r_{t,i} = \max_{\mathbf{x} \in \mathcal{A}_{t,i}} \mathbf{E}[y|\mathbf{x}] - \mathbf{E}[y_{t,i}|\mathbf{x}_{t,i}]$.

In a federated learning setting, the clients cannot directly communicate with each other, but through the central server, i.e., a star-shaped communication network. Raw data collected by each client $i \in [N]$, i.e., $\{(\mathbf{x}_{s,i}, y_{s,i})\}_{s \in [T]}$, is stored locally and cannot be shared with anyone else. Instead, the clients can only communicate the parameters of the learning algorithm, e.g., models, gradients, or sufficient statistics; and the communication cost is measured by the total number of times data being transferred across the system up to time $T$, which is denoted as $C_T$.

## 3.2 Federated Linear Bandit

Prior works have studied communication-efficient federated linear bandit [28, 6], i.e., the reward function is a linear model $y_{t,i} = \mathbf{x}_{t,i}^\top \theta_\star + \eta_{t,i}$, where $\eta_{t,i}$ denotes zero-mean sub-Gaussian noise. Consider an imaginary centralized agent that has direct access to the data of all clients, so that it can compute the global sufficient statistics $A_t = \sum_{i \in [N]} \sum_{s \in [t]} \mathbf{x}_{s,i} \mathbf{x}_{s,i}^\top, b_t = \sum_{i \in [N]} \sum_{s \in [t]} \mathbf{x}_{s,i} y_{s,i}$. Then the cumulative regret incurred by this distributed learning system can match that under a centralized setting, if all $N$ clients select arms based on the global sufficient statistics $\{A_t, b_t\}$. However, it requires $N^2 T$ communication cost for the immediate sharing of each client's update to the sufficient statistics with all other clients, which is expensive for most applications.

To ensure communication efficiency, prior works like DisLinUCB [28] let each client $i$ maintain a local copy $\{A_{t-1,i}, b_{t-1,i}\}$ for arm selection, which receives immediate local update using each newly collected data sample, i.e., $A_{t,i} = A_{t-1,i} + \mathbf{x}_{t,i} \mathbf{x}_{t,i}^\top, b_{t,i} = b_{t-1,i} + \mathbf{x}_{t,i} y_{t,i}$. Then client $i$ checks whether the event $(t - t_{\text{last}}) \log(\frac{\det A_{t,i}}{\det A_{t_{\text{last}}}}) > D$ is true, where $t_{\text{last}}$ denotes the time step of last global update. If true, a new global update is triggered, such that the server will collect all clients' local update since $t_{\text{last}}$, aggregate them to compute $\{A_t, b_t\}$, and then synchronize the local sufficient statistics of all clients, i.e., set $\{A_{t,i}, b_{t,i}\} = \{A_t, b_t\}, \forall i \in [N]$.

## 3.3 Federated Generalized Linear Bandit

In this paper, we study federated bandit learning with generalized linear models, i.e., the conditional distribution of reward $y$ given context vector $\mathbf{x}$ is drawn from the exponential family [8, 20]:

$$p_{\theta_\star}(y|\mathbf{x}) = \exp\left(\frac{y\mathbf{x}^\top \theta_\star - m(\mathbf{x}^\top \theta_\star)}{g(\tau)} + h(y, \tau)\right) \tag{1}$$

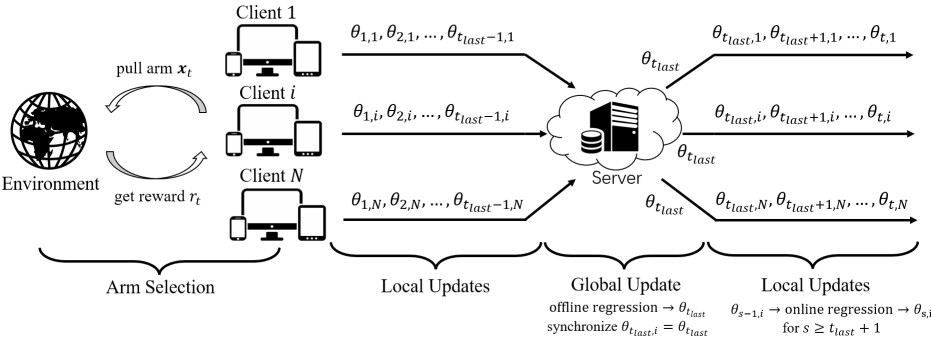

Figure 1: Illustration of FedGLB-UCB algorithm, which uses online regression for local update, i.e., immediately update each client's local model $\theta_{t,i}$ using its newly collected data sample, and uses offline regression for global update, i.e., synchronize all $N$ clients to a globally updated model $\theta_{t_{\text{last}}}$ using all the data samples collected so far.

where $\tau \in \mathbb{R}^+$ is a known scale parameter. Given a function $f : \mathbb{R} \to \mathbb{R}$, we denote its first and second derivatives by $\dot{f}$ and $\ddot{f}$, respectively. It is known that $\dot{m}(\mathbf{x}^\top\theta_\star) = \mathbb{E}[y|\mathbf{x}] := \mu(\mathbf{x}^\top\theta_\star)$, which is called the inverse link function, and $\ddot{m}(\mathbf{x}^\top\theta_\star) = \mathbb{V}(y|\mathbf{x}^\top\theta_\star)$. Based on Eq.(1), the reward $y_{t,i}$ observed by client $i$ at time $t$ can be equivalently represented as $y_{t,i} = \mu(\mathbf{x}_{t,i}^\top\theta_\star) + \eta_{t,i}$, where $\eta_{t,i}$ denotes the sub-Gaussian noise. Then we denote the negative log-likelihood of $y_{i,t}$ given $\mathbf{x}_{i,t}$ as $l(\mathbf{x}_{t,i}^\top\theta_\star, y_{t,i}) = -\log p_{\theta_\star}(y_{t,i}|\mathbf{x}_{t,i}) = -y_{t,i}\mathbf{x}_{t,i}^\top\theta_\star + m(\mathbf{x}_{t,i}^\top\theta_\star)$. In addition, we adopt the following two assumptions about the reward, which are standard for GLB [8].

**Assumption 1.** *The link function $\mu$ is continuously differentiable on $(-S, S)$, $k_\mu$-Lipschitz on $[-S, S]$, and $\inf_{z \in [-S,S]} \dot{\mu}(z) = c_\mu > 0$.*

**Assumption 2.** *$\mathbb{E}[\eta_{t,i}|\mathcal{F}_{t,i}] = 0, \forall t, i$, where $\mathcal{F}_{t,i} = \sigma\{\mathbf{x}_{t,i}, [\mathbf{x}_{s,j}, y_{s,j}]_{(s,j):s<t \cap j=i}\}$ denotes the $\sigma$-algebra generated by client $i$'s previously pulled arms and observed rewards, and $\max_{t,i}|\eta_{t,i}| \leq R_{\max}$ for some constant $R_{\max} > 0$.*

**New Challenges**    Compared with federated linear bandit discussed in Section 3.2, new challenges arise in designing a communication-efficient algorithm for federated GLB due to the absence of a closed form solution:

- *Iterative communication for global update:* compared with the global update for federated linear bandit that only requires one round of communication to share the sufficient statistics, now it takes multiple iterations of gradient aggregation to obtain converged global optimization. Moreover, as the clients collect more data samples over time during bandit learning, the required number of iterations for convergence also increases.
- *Drifting issue with local update:* during local model update, iterative optimization using only local gradient can push the updated model away from the global model, i.e., forget the knowledge gained during previous communications [15].

## 4   Methodology

In this section we propose the first algorithm for federated GLB that addresses the aforementioned challenges. We rigorously prove that it attains sub-linear rate in $T$ for both regret and communication cost. In addition, we propose and analyze different variants of our algorithm to facilitate understanding of our algorithm design.

### 4.1   FedGLB-UCB Algorithm

To ensure communication-efficient model updates for federated GLB, we propose to use online regression for local update, i.e., update each client's local model only with its newly collected data samples, and use offline regression for global update, i.e., solicit all clients' local gradients for joint model estimation. Based on the resulting sequence of *offline-and-online* model updates, the

confidence ellipsoid for $\theta_\star$ is constructed for each client to select arms using the OFUL principle. We name this algorithm Federated Generalized Linear Bandit with Upper Confidence Bound, or FedGLB-UCB for short. We illustrate its key components in Figure 1 and describe its procedures in Algorithm 1. In the following, we discuss about each component of FedGLB-UCB in details.

---

**Algorithm 1** FedGLB-UCB

---

1: **Input:** threshold $D$, regularization parameter $\lambda > 0$, $\delta \in (0, 1)$ and $c_\mu$.
2: **Initialize** $\forall i \in [N]$: $A_{0,i} = \frac{\lambda}{c_\mu}\mathbf{I} \in \mathbb{R}^{d \times d}, b_{0,i} = \mathbf{0} \in \mathbb{R}^d, \theta_{0,i} = \mathbf{0} \in \mathbb{R}^d, \Delta A_{0,i} = \mathbf{0} \in \mathbb{R}^{d \times d}$;
$A_0 = \frac{\lambda}{c_\mu}\mathbf{I} \in \mathbb{R}^{d \times d}, b_0 = \mathbf{0} \in \mathbb{R}^d, \theta_0 = \mathbf{0} \in \mathbb{R}^d, t_{\text{last}} = 0$
3: **for** $t = 1, 2, ..., T$ **do**
4:     **for** client $i = 1, 2, ..., N$ **do**
5:         Observe arm set $\mathcal{A}_{t,i}$ for client $i$
6:         Select arm $\mathbf{x}_{t,i} \in \mathcal{A}_{t,i}$ by Eq.(5), and observe reward $y_{t,i}$
7:         Update client $i$: $A_{t,i} = A_{t-1,i} + \mathbf{x}_{t,i}\mathbf{x}_{t,i}^\top, \Delta A_{t,i} = \Delta A_{t-1,i} + \mathbf{x}_{t,i}\mathbf{x}_{t,i}^\top$
8:         **if** $(t - t_{\text{last}}) \log \frac{\det(A_{t,i})}{\det(A_{t,i} - \Delta A_{t,i})} < D$ **then**
9:             **Client** $i$: perform local update $\theta_{t,i} = \text{ONS-Update}(\theta_{t-1,i}, A_{t,i}, \nabla l(\mathbf{x}_{t,i}^\top \theta_{t-1,i}, y_{t,i}))$,
$b_{t,i} = b_{t-1,i} + \mathbf{x}_{t,i}\mathbf{x}_{t,i}^\top \theta_{t-1,i}$
10:         **else**
11:             **Clients** $\forall i \in [N]$: send $\Delta A_{t,i}$ to server, and reset $\Delta A_{t,i} = \mathbf{0}$
12:             **Server**: compute $A_t = A_{t_{\text{last}}} + \sum_{i=1}^N \Delta A_{t,i}$
13:             **Server**: perform global update $\theta_t = \text{AGD-Update}(\theta_{t_{\text{last}}}, J_t)$ (see Eq.(3) for the choice of $J_t$), $b_t = b_{t_{\text{last}}} + \sum_{i=1}^N \Delta A_{t,i}\theta_t$, and set $t_{\text{last}} = t$
14:             **Clients** $\forall i \in [N]$: set $\theta_{t,i} = \theta_t, A_{t,i} = A_t, b_{t,i} = b_t$

---

● **Local update.** As mentioned earlier, iterative optimization over local dataset $\{(\mathbf{x}_{s,i}, y_{s,i})\}_{s \in [t]}$ leads to the drifting issue that pushes the updated model to the local optimum. Due to the small size of this local dataset, the confidence ellipsoid centered at the converged model has increased width, which leads to increased regret in bandit learning. However, as we will prove in Section 4.2, completely disabling local update and restricting all clients to use the previous globally updated model for arm selection is also a bad choice, because the learning system will then need more frequent global updates to adapt to the growing dataset.

To enable local update while alleviating the drifting issue, we adopt online regression in each client, such that the local model estimation $\theta_{t,i}$ is only updated for one step using the sample $(\mathbf{x}_{t,i}, y_{t,i})$ collected at time $t$. Prior works [2, 12] showed that UCB-type algorithms with online regression can attain comparable cumulative regret to the standard UCB-type algorithms [1, 20], as long as the selected online regression method guarantees logarithmic online regret. As the negative log-likelihood loss defined in Section 3.3 is exp-concave and online Newton step (ONS) is known to attain logarithmic online regret in this case [10, 12], ONS is chosen for the local update of FedGLB-UCB and its description is given in Algorithm 2. At time step $t$, after client $i$ pulls an arm $\mathbf{x}_{t,i} \in \mathcal{A}_{t,i}$ and observes the reward $y_{t,i}$, its model $\theta_{t-1,i}$ is immediately updated by the ONS update rule (line 9 in Algorithm 1), where $\nabla l(\mathbf{x}_{t,i}^\top \theta_{t-1,i}, y_{t,i})$ denotes the gradient w.r.t. $\theta_{t-1,i}$, and $A_{t,i}$ denotes the covariance matrix for client $i$ at time $t$.

---

**Algorithm 2** ONS-Update

---

1: **Input:** $\theta_{t-1,i}, A_{t,i}, \nabla l(\mathbf{x}_{t,i}^\top \theta_{t-1,i}, y_{t,i})$
2: $\theta_{t,i}' = \theta_{t-1,i} - \frac{1}{c_\mu} A_{t,i}^{-1} \nabla l(\mathbf{x}_{t,i}^\top \theta_{t-1,i}, y_{t,i})$
3: $\theta_{t,i} = \arg\min_{\theta \in \mathcal{B}_d(S)} ||\theta - \theta_{t,i}'||_{A_{t,i}}^2$
4: **Output:** $\theta_{t,i}$

---

● **Global update** The global update of FedGLB-UCB requires communication among the $N$ clients, which imposes communication cost in two aspects: 1) each global update for federated GLB requires multiple rounds of communication among $N$ clients, i.e., iterative aggregation of local gradients; and 2) global update needs to be performed for multiple times over time horizon $T$, in order to adapt to the growing dataset collected by each client during bandit learning. Consider a particular time step

$t \in [T]$ when global update happens, the distributed optimization objective is:

$$\min_{\theta \in \Theta} F_t(\theta) := \frac{1}{N} \sum_{i=1}^{N} F_{t,i}(\theta) \tag{2}$$

where $F_{t,i}(\theta) = \frac{1}{t} \sum_{s=1}^{t} l(\mathbf{x}_{s,i}^\top \theta, y_{s,i}) + \frac{\lambda}{2t} \|\theta\|_2^2$ denotes the *average* regularized negative log-likelihood loss for client $i \in [N]$, and $\lambda > 0$ denotes the regularization parameter. Based on Assumption 1, $\{F_{t,i}(\theta)\}_{i \in [N]}$ are $\frac{\lambda}{Nt}$-strongly-convex and $(k_\mu + \frac{\lambda}{Nt})$-smooth in $\theta$ (proof in Appendix A), and we denote the unique minimizer of Eq.(2) as $\hat{\theta}_t^{\text{MLE}}$. In this case, it is known that the number of communication rounds $J_t$ required to attain a specified sub-optimality $\epsilon_t$, such that $F_t(\theta) - \min_{\theta \in \Theta} F_t(\theta) \le \epsilon_t$, has a lower bound $J_t = \Omega\left(\sqrt{(k_\mu Nt)/\lambda + 1} \log \frac{1}{\epsilon_t}\right)$ [3], which means $J_t$ increases at least at the rate of $\sqrt{Nt}$. This lower bound is matched by the distributed version of accelerated gradient descent (AGD) [25]:

$$J_t \le 1 + \sqrt{(k_\mu Nt)/\lambda + 1} \log \frac{(k_\mu + \frac{2\lambda}{Nt})\|\theta^{(1)} - \hat{\theta}_t^{\text{MLE}}\|_2^2}{2\epsilon_t} \tag{3}$$

where the superscript $(i)$ denotes the $i$-th iteration of AGD.

In order to minimize the number of communication rounds in one global update, AGD is chosen as the offline regression method for FedGLB-UCB, and its description is given in Algorithm 3 (subscript $t$ is omitted for simplicity). However, other federated/distributed optimization methods can be readily used in place of AGD, as our analysis only requires the convergence result of the adopted method. We should note that $\epsilon_t$ is essential to the regret-communication trade-off during the global update at time $t$: a larger $\epsilon_t$ leads to a wider confidence ellipsoid, which increases regret, while a smaller $\epsilon_t$ requires more communication rounds $J_t$, which increases communication cost. In Section 4.2, we will discuss the proper choice of $\epsilon_t$ to attain desired trade-off between the two conflicting objectives.

---

**Algorithm 3** AGD-Update

1: **Input** : initial $\theta$, number of inner iterations $J$
2: **Initialization**: set $\theta^{(1)} = \vartheta^{(1)} = \theta$, and define the sequences $\{\upsilon_j := \frac{1+\sqrt{1+4\upsilon_{j-1}^2}}{2}\}_{j \in [J]}$ (with $\upsilon_0 = 0$), and $\{\gamma_j = \frac{1-\upsilon_j}{\upsilon_{j+1}}\}_{j \in [J]}$
3: **for** $j = 1, 2, \ldots, J$ **do**
4:  **Clients** compute and send local gradient $\{\nabla F_i(\theta^{(j)})\}_{i \in [N]}$ to the server
5:  **Server** aggregates local gradients $\nabla F(\theta^{(j)}) = \frac{1}{N} \sum_{i=1}^{N} \nabla F_i(\theta^{(j)})$, and execute the following update rule to get $\theta^{(j+1)}$:
   - $\vartheta^{(j+1)} = \theta^{(j)} - \frac{1}{k_\mu + \frac{\lambda}{Nt}} \nabla F(\theta^{(j)})$
   - $\theta^{(j+1)} = (1 - \gamma_j)\vartheta^{(j+1)} + \gamma_j \vartheta^{(j)}$
6: **Output:** $\arg\min_{\theta \in \mathcal{B}_d(S)} \|g_t(\theta^{(J+1)}) - g_t(\theta)\|_{A_t^{-1}}$

---

To reduce the total number of global updates over time horizon $T$, we adopt the event-triggered communication from [28], such that global update is triggered if the following event is true for any client $i \in [N]$ (line 8):

$$(t - t_{\text{last}}) \log \frac{\det(A_{t,i})}{\det(A_{t,i} - \Delta A_{t,i})} > D \tag{4}$$

where $\Delta A_{t,i}$ denotes client $i$'s local update to its covariance matrix since last global update at $t_{\text{last}}$, and $D > 0$ is the chosen threshold for the event-trigger. During the global update, the model estimation $\theta_{t,i}$, covariance matrix $A_{t,i}$ and vector $b_{t,i}$ for all clients $i \in [N]$ will be updated (line 11-14). We should note that the LHS of Eq.(4) is essentially an upper bound of the cumulative regret that client $i$'s locally updated model has incurred since $t_{\text{last}}$. Therefore, this event-trigger guarantees that a global update only happens when effective regret reduction is possible.

• **Arm selection** To balance exploration and exploitation during bandit learning, FedGLB-UCB uses the OFUL principle for arm selection [1], which requires the construction of a confidence ellipsoid for each client $i$. We propose a novel construction of the confidence ellipsoid based on the sequence of model updates that each client $i$ has received up to time $t$: basically, there are 1)

one global update at $t_{\text{last}}$, i.e., the joint offline regression across all clients' accumulated data till $t_{\text{last}}$: $\{(\mathbf{x}_{s,i}, y_{s,i})\}_{s\in[t_{\text{last}}], i\in[N]}$, which resets all clients' local models to $\theta_{t_{\text{last}}}$; and 2) multiple local updates from $t_{\text{last}}+1$ to $t$, i.e., the online regression on client $i$'s own data sequence $\{(\mathbf{x}_{s,i}, y_{s,i})\}_{s\in[t_{\text{last}}+1, t]}$ to get $\{\theta_{s,i}\}_{s\in[t_{\text{last}}+1, t]}$ step by step. This can be more easily understood by the illustration in Figure 1. The resulting confidence ellipsoid is centered at the ridge regression estimator $\hat{\theta}_{t,i} = A_{t,i}^{-1} b_{t,i}$ [2, 12], which is computed using the predicted rewards given by the past sequence of model updates $\{\theta_{t_{\text{last}}}\} \cup \{\theta_{s,i}\}_{s\in[t_{\text{last}}+1, t]}$ (see the update of $b_{t,i}$ in line 9 and 13 of Algorithm 1). Then at time step $t$, client $i$ selects the arm that maximizes the UCB score:

$$\mathbf{x}_{t,i} = \underset{\mathbf{x}\in\mathcal{A}_{t,i}}{\arg\max} \, \mathbf{x}^\top \hat{\theta}_{t-1,i} + \alpha_{t-1,i} ||\mathbf{x}||_{A_{t-1,i}^{-1}} \tag{5}$$

where $\alpha_{t-1,i}$ is the parameter of the confidence ellipsoid given in Lemma 4.2. Note that compared with standard federated/distributed learning where clients only need to communicate gradients for joint model estimation, in our problem, due to the time-varying arm set, it is also necessary to communicate the confidence ellipsoid among clients, i.e., $A_t \in \mathbb{R}^{d\times d}$ and $b_t \in \mathbb{R}^d$ (line 14 in Algorithm 1), as the clients need to be prepared for all possible arms $\mathbf{x} \in \mathbb{R}^d$ that may appear in future for the sake of regret minimization.

## 4.2 Theoretical Analysis

In this section, we construct the confidence ellipsoid based on the *offline-and-online* estimators described in Section 4.1. Then we analyze the cumulative regret and communication cost of FedGLB-UCB, followed by theoretical comparisons with its different variants.

• **Construction of confidence ellipsoid** Compared with prior works that convert a sequence of online regression estimators to confidence ellipsoid [2, 12], our confidence ellipsoid is built on the combination of an offline regression estimator $\theta_{t_{\text{last}}}$ for global update, and the subsequent online regression estimators $\{\theta_{s-1,i}\}_{s\in[t_{\text{last}}+1, t]}$ for local updates on each client $i$. This construction is new and requires proof techniques unique to our proposed solution. In the following, we highlight the key steps, and refer our readers to the appendix for details.

To simplify the use of notations, we assume without loss of generality that the global update at $t_{\text{last}}$ is triggered by the $N$-th client, such that no more new data will be collected at $t_{\text{last}}$, i.e., the first data sample obtained after the global update has index $t_{\text{last}}+1$. We start our construction by considering the following loss difference introduced by the global and local model updates: $\sum_{s=1}^{t_{\text{last}}} \sum_{i=1}^{N} [l(\mathbf{x}_{s,i}^\top \theta_{t_{\text{last}}}, y_{s,i}) - l(\mathbf{x}_{s,i}^\top \theta_\star, y_{s,i})] + \sum_{s=t_{\text{last}}+1}^{t} [l(\mathbf{x}_{s,i}^\top \theta_{s-1,i}, y_{s,i}) - l(\mathbf{x}_{s,i}^\top \theta_\star, y_{s,i})]$, where the first term is the loss difference between the globally updated model $\theta_{t_{\text{last}}}$ and $\theta_\star$, and the second term is between the sequence of locally updated models $\{\theta_{s-1,i}\}_{s\in[t_{\text{last}}+1, t]}$ and $\theta_\star$. This extends the definition of online regret used in the construction in [2, 12]; and due to the existence of offline regression, the obtained upper bounds in Lemma 4.1 are unique to our solution.

**Lemma 4.1** (Upper Bound of Loss Difference). *Denote the sub-optimality of the global model update procedure at time step $t_{\text{last}}$ as $\epsilon_{t_{\text{last}}}$, such that $F_{t_{\text{last}}}(\theta) - \min_{\theta\in\mathcal{B}_d(S)} F_{t_{\text{last}}}(\theta) \leq \epsilon_{t_{\text{last}}}$. Then under Assumption 1 and 2, we have*

$$\sum_{s=1}^{t_{\text{last}}} \sum_{i=1}^{N} [l(\mathbf{x}_{s,i}^\top \theta_{t_{\text{last}}}, y_{s,i}) - l(\mathbf{x}_{s,i}^\top \theta_\star, y_{s,i})] \leq B_1 \tag{6}$$

*where $B_1 = N t_{\text{last}} \epsilon_{t_{\text{last}}} + \frac{\lambda}{2} S^2$, and with probability at least $1 - \delta$,*

$$\sum_{s=t_{\text{last}}+1}^{t} [l(\mathbf{x}_{s,i}^\top \theta_{s-1,i}, y_{s,i}) - l(\mathbf{x}_{s,i}^\top \theta_\star, y_{s,i})] \leq B_2 \tag{7}$$

*where $B_2 = \frac{1}{2c_\mu} \sum_{s=t_{\text{last}}+1}^{t} \|\nabla l(\mathbf{x}_{s,i}^\top \theta_{s-1,i}, y_{s,i})\|_{A_{s,i}^{-1}}^2 + \frac{c_\mu}{2} \left[ \frac{1}{c_\mu} R_{\max} \sqrt{d \log(1 + N t_{\text{last}} c_\mu / d\lambda) + 2\log(1/\delta)} + 2 N t_{\text{last}} \sqrt{\frac{2k_\mu}{\lambda c_\mu} + \frac{2}{N t_{\text{last}} c_\mu}} \sqrt{\epsilon_{t_{\text{last}}}} + \sqrt{\frac{\lambda}{c_\mu}} S \right]^2$, respectively.*

Specifically, $B_1$ corresponds to the convergence of the offline (distributed) optimization in previous global update; $B_2$ is essentially the online regret upper bound of ONS, with the major difference that it is initialized using the globally updated model $\theta_{t_{\text{last}}}$, instead of an arbitrary model as in standard

ONS. Then due to the $c_\mu$-strongly-convexity of $l(z, y)$ w.r.t. $z$, i.e., $l(\mathbf{x}_s^\top \theta, y_s) - l(\mathbf{x}_s^\top \theta_\star, y_s) \geq \left[\mu(\mathbf{x}_s^\top \theta_\star) - y_s\right]\mathbf{x}_s^\top(\theta - \theta_\star) + \frac{c_\mu}{2}\left[\mathbf{x}_s^\top(\theta - \theta_\star)\right]^2$, and by rearranging terms in Eq.(6) and Eq.(7), we have: $\sum_{s=1}^{t_{last}} \sum_{i=1}^{N} \left[\mathbf{x}_{s,i}^\top(\theta_{t_{last}} - \theta_\star)\right]^2 \leq \frac{2}{c_\mu}B_1 + \frac{2}{c_\mu}\sum_{s=1}^{t_{last}}\sum_{i=1}^{N}\eta_{s,i}\mathbf{x}_{s,i}^\top(\theta_{t_{last}} - \theta_\star)$, and $\sum_{s=t_{last}+1}^{t}\left[\mathbf{x}_{s,i}^\top(\theta_{s-1,i} - \theta_\star)\right]^2 \leq \frac{2}{c_\mu}B_2 + \frac{2}{c_\mu}\sum_{s=t_{last}+1}^{t}\eta_{s,i}\mathbf{x}_{s,i}^\top(\theta_{s-1,i} - \theta_\star)$, whose LHS is quadratic in $\theta_\star$. To further upper bound the RHS, we should note that the term $\frac{2}{c_\mu}\sum_{s=t_{last}+1}^{t}\eta_{s,i}\mathbf{x}_{s,i}^\top(\theta_{s-1,i} - \theta_\star)$ is standard in [2, 12] as $\mathbf{x}_{s,i}^\top(\theta_{s-1,i} - \theta_\star)$ is $\mathcal{F}_{s,i}$-measurable for online estimator $\theta_{s-1,i}$. However, this is not true for the term $\frac{2}{c_\mu}\sum_{s=1}^{t_{last}}\sum_{i=1}^{N}\eta_{s,i}\mathbf{x}_{s,i}^\top(\theta_{t_{last}} - \theta_\star)$ as the offline regression estimator $\theta_{t_{last}}$ depends on all data samples collected till $t_{last}$; and thus we have to develop a different approach to bound it. This leads to Lemma 4.2 below, which provides the confidence ellipsoid for $\theta_\star$.

**Lemma 4.2** (Confidence Ellipsoid of FedGLB-UCB). *With probability at least $1 - 2\delta$, for all $t \in [T], i \in [N]$,*

$$\|\hat{\theta}_{t,i} - \theta_\star\|_{A_{t,i}}^2 \leq \beta_{t,i} + \frac{\lambda}{c_\mu}S^2 - \|z_{t,i}\|_2^2 + \hat{\theta}_{t,i}^\top b_{t,i} := \alpha_{t,i}^2$$

*where $z_{t,i}$ denotes the vector of predicted rewards $[\mathbf{x}_{1,1}^\top \theta_{t_{last}}, \mathbf{x}_{1,2}^\top \theta_{t_{last}}, \dots, \mathbf{x}_{t_{last},N-1}^\top \theta_{t_{last}}, \mathbf{x}_{t_{last},N}^\top \theta_{t_{last}}, \mathbf{x}_{t_{last}+1,i}^\top \theta_{t_{last},i}, \mathbf{x}_{t_{last}+2,i}^\top \theta_{t_{last}+1,i}, \dots, \mathbf{x}_{t,i}^\top \theta_{t-1,i}]^\top$, and $\beta_{t,i} = \frac{8R_{max}^2}{c_\mu^2}\log\left(\frac{1}{\delta}\sqrt{\det(I + \sum_{s=1}^{t_{last}}\sum_{i=1}^{N}\mathbf{x}_{s,i}\mathbf{x}_{s,i}^\top)}\right) + B_1 + \frac{4R_{max}}{c_\mu}\sqrt{2\log\left(\frac{1}{\delta}\sqrt{\det(I + \sum_{s=1}^{t_{last}}\sum_{i=1}^{N}\mathbf{x}_{s,i}\mathbf{x}_{s,i}^\top)}\right)}\left(\|\theta_{t_{last}}\|_2 + \|\theta_\star\|_2 + \sqrt{B_1}\right) + \frac{4B_2}{c_\mu} + \frac{8R_{max}^2}{c_\mu^2}\log\left(\frac{N}{\delta}\sqrt{4 + \frac{8}{c_\mu}B_2 + \frac{64R_{max}^4}{c_\mu^4 \cdot 4\delta^2}}\right) + 1.$*

**• Regret and communication cost** From Lemma 4.2, we can see that $\alpha_{t,i}$ grows at a rate of $Nt_{last}\sqrt{\epsilon_{t_{last}}}$ through its dependence on the $B_2$ term. To make sure the growth rate of $\alpha_{t,i}$ matches that in standard GLB algorithms [20, 12], we set $\epsilon_{t_{last}} = \frac{1}{N^2 t_{last}^2}$, which leads to the following corollary.

**Corollary 4.2.1** (Order of $\beta_{t,i}$). *With $\epsilon_{t_{last}} = \frac{1}{N^2 t_{last}^2}$, $\beta_{t,i} = O\left(\frac{d \log NT}{c_\mu^2}[k_\mu^2 + R_{max}^2]\right)$.*

Then using a similar argument as the proof for Theorem 4 of [28], we obtain the following upper bounds on $R_T$ and $C_T$ for FedGLB-UCB (proof in Appendix D).

**Theorem 4.3** (Regret and Communication Cost Upper Bound of FedGLB-UCB). *Under Assumption 1, 2, and by setting $\epsilon_t = \frac{1}{N^2 t^2}, \forall t$ and $D = \frac{T}{Nd\log(NT)}$, the cumulative regret $R_T$ has upper bound*

$$R_T = O\left(\frac{k_\mu(k_\mu + R_{max})}{c_\mu}d\sqrt{NT}\log(NT/\delta)\right),$$

*with probability at least $1 - 2\delta$. The corresponding communication cost $C_T$[1] has upper bound*

$$C_T = O\left(dN^2\sqrt{T}\log^2(NT)\right).$$

Theorem 4.3 shows that FedGLB-UCB recovers the standard $O\left(d\sqrt{NT}\log(NT)\right)$ rate in regret as in the centralized setting, while only incurring a communication cost that is sub-linear in $T$. Note that, to obtain $O\left(d\sqrt{NT}\log(NT)\right)$ regret for federated linear bandit, the DisLinUCB algorithm incurs a communication cost of $O(dN^{1.5}\log(NT))$ [28], which is smaller than that of FedGLB-UCB by a factor of $\sqrt{NT}\log(NT)$. As the frequency of global updates is the same for both algorithms (due to their use of the same event-trigger), this additional communication cost is caused by the iterative optimization procedure for the global update, which is required for GLB model estimation. Moreover, as we mentioned in Section 4.1, there is not much room for improvement here as the use of AGD already matches the lower bound up to a logarithmic factor.

To facilitate the understanding of our algorithm design and investigate the impact of different components of FedGLB-UCB on its regret and communication efficiency trade-off, we propose

---

[1]This is measured by the *total number of times* data is transferred. Some works [28] measure $C_T$ by the *total number of scalars* transferred, in which case, we have $C_T = O\left(d^3 N^{1.5}\log(NT) + d^2 N^2 T^{0.5}\log^2(NT)\right)$.

Table 1: Comparison between FedGLB-UCB and its variants with different design choices.

| Global Upd. | Local Upd. | Setting | $R_T$ | $C_T$ |
|---|---|---|---|---|
| AGD | ONS | $D = \frac{T}{Nd\log(NT)}$ | $\frac{k_\mu(k_\mu+R_{\max})}{c_\mu}d\sqrt{NT}\log(NT)$ | $dN^2\sqrt{T}\log^2(NT)$ |
| AGD | no update | $B = \sqrt{NT}$ | $\frac{k_\mu R_{\max}}{c_\mu}d\sqrt{NT}\log(NT)$ | $N^2T\log(NT)$ |
| AGD | ONS | $B = d^2N\log(NT)$ | $\frac{k_\mu(k_\mu+R_{\max})}{c_\mu}d\sqrt{NT}\log(NT)\log(T)$ | $d^2N^{2.5}\sqrt{T}\log^2(NT)$ |
| ONS | ONS | $B = \sqrt{NT}$ | $\frac{k_\mu(k_\mu+R_{max})}{c_\mu}d(NT)^{3/4}\log(NT)$ | $N^{1.5}\sqrt{T}$ |

and analyze three variants, which are also of independent interest, and report the results in Table 1. Detailed descriptions, as well as proof for these results can be found in Appendix E. Note that all three variants perform global update according to a fixed schedule $\mathcal{S} = \{t_1 := \lfloor\frac{T}{B}\rfloor, t_2 := 2\lfloor\frac{T}{B}\rfloor, \ldots, t_B := B\lfloor\frac{T}{B}\rfloor\}$, where $B$ denotes the total number of global updates specified in advance to trade-off between $R_T$ and $C_T$, and these variants differ in their global and local update strategies. This comparison demonstrates that our solution is proven to achieve a better regret-communication trade-off against these reasonable alternatives. For example, when using standard federated learning methods (which assume fixed dataset) for streaming data in real-world applications, it is a common practice to set some fixed schedule to periodically retrain the global model to fit the new dataset, and FedGLB-UCB$_1$ implements such behaviors. The design of FedGLB-UCB$_3$ is motivated by distributed online convex optimization that also deals with streaming data in a distributed setting.

## 5   Experiments

We performed extensive empirical evaluations of FedGLB-UCB on both synthetic and real-world datasets, and the results (averaged over 10 runs) are reported in Figure 2. We included the three variants of FedGLB-UCB (listed in Table 1), One-UCB-GLM, N-UCB-GLM [20] and N-ONS-GLM [12] as baselines, where One-UCB-GLM learns a shared bandit model across all clients, and N-UCB-GLM and N-ONS-GLM learn a separated bandit model for each client with no communication. Additional results and discussions about experiments can be found in Appendix F.

• **Synthetic Dataset** We simulated the federated GLB setting defined in Section 3.3, with $T = 2000, N = 200, d = 10, S = 1, \mathcal{A}_t$ ($K = 25$) uniformly sampled from a $\ell_2$ unit sphere, and reward $y_{t,,i} \sim \text{Bernoulli}(\mu(\mathbf{x}_{t,,i}^\top \theta_\star))$, with $\mu(z) = (1 + \exp(-z))^{-1}$. To compare the algorithms' $R_T$ and $C_T$ under different trade-off settings, we run FedGLB-UCB with different threshold value $D$ (logarithmically spaced between $10^{-1}$ and $10^3$) and its variants with different number of global updates $B$. Note that each dot in the result figure illustrates the $C_T$ (x-axis) and $R_T$ (y-axis) that a particular instance of FedGLB-UCB or its variants obtained by time $T$, and the corresponding value for $D$ or $B$ is labeled next to the dot. $R_T$ of One-UCB-GLM is illustrated as the red horizontal line, and $R_T$ of N-UCB-GLM and N-ONS-GLM are labeled on the top of the figure. We can observe that for FedGLB-UCB and its variants, $R_T$ decreases as $C_T$ increases, interpolating between the two extreme cases: independently learned bandit models by N-UCB-GLM, N-ONS-GLM; and the jointly learned bandit model by One-UCB-GLM. FedGLB-UCB significantly reduces $C_T$, while attaining low $R_T$, i.e., its regret is even comparable with One-UCB-GLM that requires at least $C_T = N^2T$ ($8 \times 10^7$ in this simulation) for gradient aggregation at each time step.

• **Real-world Dataset** The results above demonstrate the effectiveness of FedGLB-UCB when data is generated by a well-specified generalized linear model. To evaluate its performance in a more challenging and practical scenario, we performed experiments using real-world datasets: CoverType, MagicTelescope and Mushroom from the UCI Machine Learning Repository [5]. To convert them to contextual bandit problems, we pre-processed these datasets following the steps in prior works [8], with $T = 2000$ and $N = 20$. Moreover, to demonstrate the advantage of GLB over linear model, we included DisLinUCB [28] as an additional baseline. Since the parameters being communicated in DisLinUCB and FedGLB-UCB are different, to ensure a fair comparison of $C_T$ in this experiment, we measure communication cost (x-axis) by the number of integers or real numbers transferred across the learning system (instead of the frequency of communications). Note that DisLinUCB has no $C_T \geq 3 \times 10^6$ in Figure 2 because its global update is already happening in every round and cannot be increased further. As mentioned earlier, due to the difference in messages being sent, the communication in DisLinUCB's *per global update* is much smaller than that in FedGLB-UCB. However, because linear models failed to capture the complicated reward mappings in these three

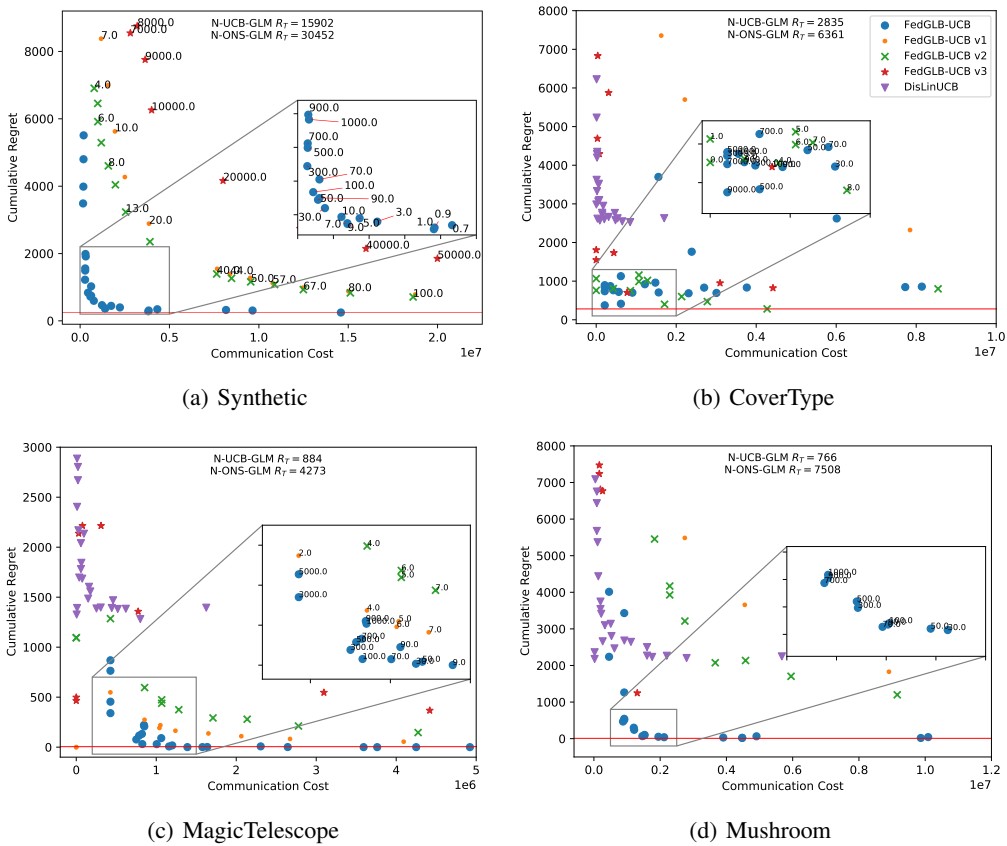

(a) Synthetic            (b) CoverType

(c) MagicTelescope       (d) Mushroom

Figure 2: Experiment results on synthetic and real world datasets.

datasets, we can see that DisLinUCB is clearly outperformed by FedGLB-UCB and its variants. This shows that, by offering a larger variety of modeling choices, e.g., linear, Poisson, logistic regression, etc., FedGLB-UCB has more potential in dealing with the complicated data in real-world applications.

## Conclusion

In this paper, we take the first step to address the new challenges in communication efficient federated bandit learning beyond linear models, where closed-form solutions do not exist, and propose a solution framework for federated GLB that employs online regression for local update and offline regression for global update. For arm selection, we propose a novel confidence ellipsoid construction based on the sequence of *offline-and-online* model estimations. We rigorously prove that the proposed algorithm attains sub-linear rate for both regret and communication cost, and also analyze the impact of each component of our algorithm via theoretical comparison with different variants. In addition, extensive empirical evaluations are performed to validate the effectiveness of our algorithm.

An important further direction of this work is the lower bound analysis for the communication cost, analogous to the communication lower bound for standard distributed optimization by Arjevani and Shamir [3]. Moreover, in our algorithm, clients' locally updated models are not utilized for global model update, so that another interesting direction is to investigate whether using such knowledge, e.g., by model aggregation, can further improve communication efficiency.

## Acknowledgement

This work is supported by NSF grants IIS-2213700, IIS-2128019 and IIS-1838615.

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
