# OpenReview forum: "Communication Efficient Federated Learning for Generalized Linear Bandits"
_NeurIPS.cc/2022/Conference — NeurIPS 2022 Accept_

### Official Review · Reviewer_vtVx · 2022-07-10

**Rating:** 5
**Confidence:** 4
**Soundness:** 3 good
**Presentation:** 2 fair
**Contribution:** 3 good

**Summary:**

This paper considers federated learning for generalized linear bandits. The key difference compared to federated linear bandits is that it requires an iterative process for the global update rather than relying on sufficient statistics. The authors propose an efficient algorithm that is able to achieve a trade-off between communication and regret. Some variants of the proposed algorithm are also studied. Experiments are also conducted to corroborate the theoretical results.

**Questions:**

The reviewer has the following concerns.
- The first one is about the technical contributions. It seems to me that this paper is a simple combination of existing techniques. That is, online-to-confidence bound + distributed optimization. The authors might need to highlight the new technical challenges in this paper.
- The authors point out the issue of filtration in existing work (e.g., Lemma H.1 [28]) in the appendix. But it seems to the reviewer that the current explanation is not clear. Can the authors elaborate on this possible gap?
- The presentation for the main part, especially Section 4.2 needs improvement. The current presentation makes readers difficult to appreciate new insights in this paper compared to existing work [2,12]. The authors can try to only give an informal version of the lemma and results by only highlighting the key terms and discarding all other similar terms in [2,12].
- Another question is about the measure of the communication overhead. It would also be important to measure the cost in terms of real numbers rather than the communication times.
- Also, the current communication is a synchronized one, i.e., when one local client triggered it, all the clients need to communicate. This could be impractical. Can the authors comment on how to handle an asynchronous communication scheme?

**Limitations:**

Yes.

**Strengths And Weaknesses:**

**Strengths**
- A first study on the new problem
- Both theoretical results and empirical experiments

**Weaknesses**
- Technical contributions are limited
- The presentation may be improved

---

> ### Author Response · Authors · 2022-08-02
> **Response to Reviewer vtVx [Part 1/2]**
>
> We will improve the presentation of Section 4.2 and highlight the insights in the revised version.
>
> **[Q1]** It seems to me that this paper is a simple combination of existing techniques. That is, online-to-confidence bound + distributed optimization. The authors might need to highlight the new technical challenges in this paper.**
>
> **[R1]** Existing federated bandit learning solutions only work with linear models, and cannot be extended to afford non-linear models like generalized linear models (GLM) or neural networks, which are essential to capture complicated reward mappings in practice. We took the first step in this direction, by solving this challenging problem via online and offline distributed regressions and carefully designing the corresponding confidence ellipsoid for efficient exploration.
>
> We argue that this reduction itself is novel in bandit literature, and also brings several new technical challenges. As mentioned in Section 4.2, initializing online regression with offline regression introduces dependencies that require proof techniques unique to this solution, and the selection of offline regression method, as well as its stopping criterion, are carefully chosen to balance regret and communication. For example, our analysis indicates that FedAvg, despite being popularly used in practice, leads to linear communication cost in the FedGLB problem due to its sub-optimal convergence rate. We also want to emphasize that our solution is proven to achieve a better regret-communication trade-off against many other alternative choices (shown in Table 1), which also demonstrates our solution is not a straightforward combination.
>
> For example, when using standard federated learning methods (which assume fixed dataset) for streaming data in real-world applications, it is a common practice to set some fixed schedule to periodically retrain the global model to fit the new dataset, and FedGLB-UCB$\_{1}$ implements such behaviors. And FedGLB-UCB$\_{3}$ stems from distributed online convex optimization that also deals with streaming data in a distributed setting. The insights from our theoretical results and proof techniques are of independent interest, and can facilitate future research in this direction.
>
> **[Q2]** The authors point out the issue of filtration in existing work (e.g., Lemma H.1 [28]) in the appendix. But it seems to the reviewer that the current explanation is not clear. Can the authors elaborate on this possible gap?
>
> **[R2]** The self-normalized bound in Theorem 2 of [1] depends on the corresponding filtration, which is not stated clearly in the proof of Lemma H.1 [28]. In particular, when claiming Theorem 2 of [1] can be used to show that the true parameter $\theta_{\star}$ lies in the confidence ellipsoid $C_{t,i}$ of some client $i$ for all $t$ with high probability, we need to be careful about what the corresponding filtration is, as $C_{t,i}$ contains data from both client $i$ and data from other clients that are received from the synchronization step.
>
> Denote the sequence of time indices where the global synchronization happens as $t_{p}$ for $p \in [P]$ for some $P>0$. Then we look at how the data on some particular client $i$ changes. When $t \in (0, t_{1})$, i.e., before any communication, client $i$ collects data samples one-by-one via sequential interactions with the environment, which is the same as in the centralized setting. Then at the global synchronization step $t=t_{1}$, client $i$ will receive a batch of $(N-1)t_{1}$ data points (in the form of sufficient statistics) from other clients. This “sequence of single data sample" and then “batch of data samples" procedure repeats until the end of time horizon $T$. The question now is how to construct the filtration such that Theorem 2 of [1] can be applied.
>
> In the filtration of Theorem 2 of [1], each sigma algebra only contains one new data point than the previous one. However, as we have seen above, at $t=t_{p}$, a total number of $(N-1)(t_{p}-t_{p-1})$ data points comes in a batch, which is controlled by the event-trigger (which introduces dependency on all the data points up to $t_{p}$). In this case, we should modify the filtration accordingly, such that the sigma algebra at time step $t_{p}$ contains a batch of new data. Otherwise, we encounter the problem as mentioned in line 498 of our paper.

---

> ### Author Response · Authors · 2022-08-02
> **Response to Reviewer vtVx [Part 2/2]**
>
> **[Q3]** Another question is about the measure of the communication overhead. It would also be important to measure the cost in terms of real numbers rather than the communication times.
>
> **[R3]** The communication cost in terms of the real numbers is essentially a scaled version (by $d^{2}$) of our current result, and we will make this clearer in the revised version.
>
> Specifically, in Theorem 4.3, the total number of global updates required by FedGLB-UCB is $O\Big(d N^{1.5} \log(NT)\Big)$, which is the same as DisLinUCB. For each global update, the clients will first share their local updates of the $A$ matrix (line 11 of Algorithm 1), whose size is $d \times d$. Then a total number of $O(\sqrt{NT} \log(NT) )$ iterations of AGD update (line 13 of Algorithm 1) will be performed, where the local gradients and models of size $d$ will be shared. Therefore, in terms of real numbers, the communication cost is $O\Big(d^{3} N^{1.5} \log(NT) + d^{2} N^{2} T^{0.5} \log^{2}(NT)  \Big)$.
>
> **[Q4]** Also, the current communication is a synchronized one, i.e., when one local client triggered it, all the clients need to communicate. This could be impractical. Can the authors comment on how to handle an asynchronous communication scheme?
>
> **[R4]** In an asynchronous communication scheme for FedGLB, the communication between each client and the server will be controlled by their own event-trigger independently from the other clients, which is similar to [17]. However, a key challenge in our problem is that in this case only the gradient of the active client is available during global update, which causes the drifting issue of global model update.
> Our proposed variant FedGLB-UCB_{3} (see Appendix F.3 for more details) can address this issue, since its global update uses (batched) ONS as well. However, as we have shown in Table 1, this leads to the sub-optimal $O((NT)^{3/4})$ regret.
>
> A more promising direction would be applying some weighted model aggregation, which is commonly used in existing works of asynchronous federated learning to mitigate the effects of stale local models [A2]. However, we should note that these works assume deterministic datasets, i.e., the stale local models are trained using the same dataset throughout the learning process, while in our case, the dataset increases in size as the algorithm interacts with the environment on the fly. Therefore, additional assumptions about the context distribution will be needed to make these model aggregation methods work.
>
> [A2] Chen, Y., Sun, X. and Jin, Y., 2019. Communication-efficient federated deep learning with layerwise asynchronous model update and temporally weighted aggregation. IEEE transactions on neural networks and learning systems, 31(10), pp.4229-4238.

---

> ### Author Response · Authors · 2022-08-09
> **Response to Reviewer vtVx**
>
> We appreciate the reviewer's suggestion on our presentation in Section 4.2, and we have added more discussions to highlight our solution’s technical novelty compared with prior works in the revised version. We hope this makes the insights clearer.
>
> If the reviewer was suggesting to have some informal notion of our theoretical results, such as using the big O notation for the definition of $\beta_{t,i}$ in line 280-282, our Corollary 4.2.1 meets the need. However, if the reviewer is looking for something else, we would like to have some clarification on the request so as to better proceed.

---

### Official Review · Reviewer_btTD · 2022-07-11

**Rating:** 6
**Confidence:** 4
**Soundness:** 3 good
**Presentation:** 3 good
**Contribution:** 3 good

**Summary:**

This paper studies the generalized linear bandit in the federated setting. In contrast to prior work on federated/distributed bandits that consider the linear bandit setting, the generalized linear bandit is more challenging since the lack of a closed-form solution prevents naively broadcasting sufficient statistics in the generalized linear bandit. The authors propose a novel algorithm $\texttt{FedGLB-UCB}$ for the generalized linear bandit in the federated setting that combines offline and online regression in order to balance communication and computation. The authors demonstrate that $\texttt{FedGLB-UCB}$ obtains competitive regret with a moderate communication budget, and also demonstrate its efficiency on a synthetic and real-world benchmark.

**Questions:**

Please see above for questions.

**Limitations:**

The authors have discussed their limitations in the conclusion, however it would be great if they could provide more substantial remarks in those directions.

**Strengths And Weaknesses:**

Strengths:
- Federated bandits are an upcoming area of interest within the bandit community, and hence the topic of this paper is very relevant to that community.
- The authors are correct in that most prior work focuses on linear bandits in the federated setting, and extending those algorithms to the generalized linear bandit is non-trivial and hence this is an important challenge.
- The paper is well-written and easy to understand. The central algorithm design components are easy to grasp.
- The presented algorithm is no-regret with efficient communication and works well on benchmarks, outperforming prior work.

Weaknesses:
- The authors do not discuss optimality within the problem setting: what is the optimal level of communication for no-regret learning in federated generalized linear bandits?
- It appears that the suboptimality (e.g., compared to the linear bandit) arises from the global update communication. Do the authors have any concrete remarks regarding this? The update does nullify the value in local updates (as pointed out by the authors in the conclusion as well), are there any assumptions that can allow reusing local updates in the global aggregation step?

---

> ### Author Response · Authors · 2022-08-02
> **Response to Reviewer btTD**
>
> **[Q1]** The authors do not discuss optimality within the problem setting: what is the optimal level of communication for no-regret learning in federated generalized linear bandits?
>
> **[R1]** Please find our common response for the discussion about the communication lower bound.
> Moreover, we want to clarify that, “no-regret” in the sense that the cumulative regret is sub-linear in $T$, is achievable even without communication. As mentioned in **case 1** of our common response, by running standard GLB algorithms like GLM-UCB on each client independently, the regret is already $O(N\sqrt{T})$. Therefore, the main goal of our FedGLB algorithms is to obtain $O(\sqrt{NT})$ regret, i.e., sublinear in **both** $N$ and $T$. As mentioned in the common response, the only existing lower bound result states that in order to have regret smaller than $O(N\sqrt{T})$, an $\Omega(N)$ communication is necessary, but there lacks similar result stating the communication lower bound to obtain $O(\sqrt{NT})$ regret.
>
> **[Q2]** It appears that the suboptimality (e.g., compared to the linear bandit) arises from the global update communication. Do the authors have any concrete remarks regarding this? The update does nullify the value in local updates (as pointed out by the authors in the conclusion as well), are there any assumptions that can allow reusing local updates in the global aggregation step?
>
> **[R2]** Since our FedGLB-UCB has the same number of global updates as DisLinUCB, the $O(\sqrt{NT} \log(NT))$ gap compared with federated linear bandits is due to our iterative optimization at each global update, whose root cause is GLM does not have a closed form estimation.
>
> As mentioned in line 201 of our paper, the upper bound for the number of iterations in each global update in our solution already matches the lower bound for the standard distributed optimization problem. However, as indicated in our conclusion, we may be able to further reduce the number of iterations required, since, unlike standard distributed optimization problems, we have access to both the previous global model and the $N$ new local models obtained by locally updating the previous global model on each client. Intuitively, if the data distribution on the $N$ clients are similar, averaging the $N$ local models should put us to a closer position to the new global optimum, and thus speed up convergence. However, additional assumptions are needed about data distribution (e.g., the arm pool across clients), which sounds less desirable in bandit research (e.g., typically no assumptions should be made on this aspect). We leave this as a possible extension of our future work.

---

> > ### Comment · Reviewer_btTD · 2022-08-08
> > **Thanks for the response**
> >
> > Thank you to the authors for addressing my concerns. While I will maintain my score of 6, I am in favor of the paper being accepted with the changes and discussion posted during the review process.

---

### Official Review · Reviewer_Uv78 · 2022-07-11

**Rating:** 6
**Confidence:** 4
**Soundness:** 3 good
**Presentation:** 3 good
**Contribution:** 3 good

**Summary:**

This paper studies the federated generalized linear bandits problem where each client faces a generalized linear bandits model. The parameters and the link functions across clients are the same. The learning objective is to minimize the cumulative regret of those clients. The paper proposes an algorithm called FedGLB-UCB with regret bound $O(d\sqrt{T})$ and communication cost at most $O(dN^2\sqrt{T})$, where $d$ is the dimension of parameter space, $T$ is the time horizon, and $N$ is the number of clients. Experiments are performed on both synthetic and real world dataset.

**Questions:**

FedGLB-UCB adopts online regression for local update. While this reduces the computation complexity in each round, it may lead to larger estimation error. Is it possible to relace online regression with offline regression in each round for local update? Meanwhile, the global update takes many iterations of communications for each update. If the local update is already offline, can the computation complexity (hence the communication cost) for global update be reduced?

As mentioned above, the communication cost scales in $O(\sqrt{T})$ other than $\log(T)$. Is it possible to further reduce the communication cost to $O(\log(T))$ under federated generalized linear bandits? The scaling in $N$ is also not promising, as it increases as $N$ increases, which limits the scalability of the algorithm. Can this be reduced as well?

In Li, et al, (2017), it assumes that the link function is strictly increasing. Does this paper have similar assumption?

Is there applicable lower bound on regret for this problem?


**Limitations:**

The paper discusses limitations of current analysis and potential future directions.

**Strengths And Weaknesses:**

The GLB setting introduces new challenges in the federated setting, since the estimation of the parameter does not have closed form and should be iteratively updated. This imposes challenges for efficient communication, and causes drifting issues at local updates.

The proposed FedGLB-UCB algorithm addresses such challenges by combining online local updating and offline regression for global update. It rigorously proves that  FedGLB-UCB achieves the same learning regret order as in the centralized setting, with a sublinear communication cost. The paper is in general well written and easy to follow.

Although  the communication cost is sub-linear, the order of $O(N^2\sqrt{T})$ still seems a little bit large, since federated linear bandits can achieve $O(N\log(T))$.

---

> ### Author Response · Authors · 2022-08-02
> **Response to Reviewer Uv78 [Part 1/2]**
>
> **[Q1]** Although the communication cost is sub-linear, the order of $O(N^{2}\sqrt{T})$ still seems a little bit large, since federated linear bandits can achieve $O(N \log⁡(T))$
>
> **[R1]** First, we want to clarify that to achieve $O(d\sqrt{NT}\log(NT))$ regret, the existing algorithms for federated linear bandits, e.g. DisLinUCB [28] requires $O(d N^{1.5}\log(NT))$ communication cost and AsyncLinUCB [17, A1] requires $O(d N^{2}\log(NT))$ communication cost, instead of $O(d N log(NT))$.
>
> As mentioned in line 296 of our paper, compared with DisLinUCB, the additional $O(\sqrt{NT} log(NT))$ factor in the communication cost of our FedGLB-UCB algorithm results from the iterative optimization during each global update, which is hard, if not impossible, to circumvent since the objective function for GLB has no closed-form solution (see our common response about the lower bound).
>
> **[Q2]** FedGLB-UCB adopts online regression for local update. While this reduces the computation complexity in each round, it may lead to larger estimation error. Is it possible to replace online regression with offline regression in each round for local update? Meanwhile, the global update takes many iterations of communications for each update. If the local update is already offline, can the computation complexity (hence the communication cost) for global update be reduced?
>
> **[R2]** As mentioned in line 155 of our paper, during local update, each client only has access to its local data, which is required by the setting of federated learning. Thus, offline regression (multiple iterations using the gradient of the local data samples) will cause the drifting issue, i.e., it makes the client forget the global information obtained in previous communications, and hence lead to larger regret. This is why an online regression (one iteration using only the gradient of the newly collected data sample) is adopted, i.e., it slightly adjusts the local model to fit the newly collected data on each client, while avoiding drifting too far away from the previous global model with provable guarantees.
>
> Moreover, we want to clarify that the purpose of local update is to utilize newly collected data to help reduce the regret incurred by each client in the current epoch. It cannot help speed up the convergence of global updates. This is because local models are not used during the global model update, as shown in line 13 of Algorithm 1 and discussed in our conclusion section.
> However, as we mentioned in the conclusion and suggested by the reviewer, utilizing the local models for global update is a potential direction to improve the convergence and thus reduce communication cost. Some additional assumptions on the context distributions of the clients would be needed to make this solution viable. For example, we may perform model aggregation among the clients’ local models to better initialize the global update, which however needs additional assumptions on the distribution of candidate arm pools in each client.
>
> **[Q3]** Is it possible to further reduce the communication cost to O(log⁡(T)) under federated generalized linear bandits? The scaling in N is also not promising, as it increases as N increases, which limits the scalability of the algorithm. Can this be reduced as well?
>
> **[R3]** The scaling in N cannot be completely removed. As mentioned in our common response about the communication lower bound, an $\Omega(N)$ communication cost is inevitable in order to attain a non-trivial regret, i.e., smaller than the $O(N\sqrt{T})$ regret attained by running $N$ instances of GLM-UCB separately on each client with no communication.
>
> Moreover, as mentioned in our response to Q1, our $O(\sqrt{T})$ scaling is due to the iterative optimization during each global update. In comparison, an $O(\log⁡(T))$ scaling is possible for distributed linear bandit algorithms, because only one round of communication is needed for each global update thanks to its closed-form solution. As mentioned in our common response about the communication cost lower bound, we hypothesize that an $\Omega(\sqrt{T})$ scaling is also inevitable for the FedGLB problem.
>
> **[Q4]** In Li, et al, (2017), it assumes that the link function is strictly increasing. Does this paper have similar assumption?
>
> **[R4]** Yes. As shown in Assumption 1, we adopt the standard assumption about the link function in previous works of generalized linear bandits [8,12], i.e., the first-order derivative of the link function is lower bounded by a positive constant. This is not a strong assumption, since it holds true not only for canonical GLM, but also non-canonical GLM like the probit model for binary reward.

---

> ### Author Response · Authors · 2022-08-02
> **Response to Reviewer Uv78 [Part 2/2]**
>
> **[Q5]** Is there an applicable lower bound on regret for this problem?
>
> **[R5]** As mentioned in our common response about the communication lower bound, there is always a trade-off between regret and communication in federated bandit problems, so that we need to discuss regret lower bound under different constraints on communication.
> With no constraint on communication, i.e., the clients communicate in each round, the regret lower bound of FedGLB is the same as the regret lower bound of centralized GLB, which is $\Omega(\sqrt{NT})$, and our proposed FedGLB-UCB matches this lower bound (up to a logarithmic factor), i.e., it attains near-optimal regret.
>
> On the other hand, as discussed in our common response, another regret lower bound states that if the communication cost is smaller than $O(N)$, the regret of a FedGLB algorithm is at least $\Omega(N\sqrt{T})$, i.e., no better than running individualized bandit models for each client.

---

### Author Response · Authors · 2022-08-02
**Common Response to All Reviewers**

We thank all reviewers for the constructive comments. In the following, we first respond to the shared question about the communication lower bound and then answer specific questions and comments from each reviewer.

### Communication Lower Bound

We should note that, for federated bandit problems, the communication lower bound is only meaningful under different constraints/requirements on the regret. The following two extreme cases of regret are of particular interests:
1. Run $N$ instances of an optimal bandit algorithm, e.g., LinUCB for linear bandit and GLM-UCB for GLB, separately on each client with no communication, which leads to $O(N\sqrt{T})$ regret and $0$ communication cost;

2. Run one instance of the optimal bandit algorithm over the data of all $N$ clients, which leads to $O(\sqrt{NT})$ regret and communication cost linear in $NT$.

The regret upper bound in **case 2** is optimal, since it already matches the lower bound for a centralized bandit problem of length $NT$. Therefore, the goal of federated bandit algorithms is to attain the optimal regret of $O(\sqrt{NT})$, while having a communication cost sub-linear in $T$.

The main contribution of our paper is to propose the first algorithm that achieves such a goal for the FedGLB problem. The lower bound analysis for the communication cost of federated bandits is highly non-trivial, and still remains an open problem. As mentioned in our conclusion, this is an important future direction.

To the best of our knowledge, the only communication lower bound result available is for **case 1**. It was originally provided for the context-free setting (Theorem 2 of [28]), but as shown in a recent work (Theorem 5.3 of [A1]), it also applies to the linear setting (and thus applies to FedGLB as well). It states that, in order to have smaller regret than $O(N\sqrt{T})$ in **case 1**, an $\Omega(N)$ communication cost is necessary.

So far, there is no useful result for **case 2**, i.e., the communication lower bound to obtain $O(\sqrt{NT})$ regret; and none of the existing federated linear bandit algorithms [17, 28, A1] can close the gap with the $\Omega(N)$ communication lower bound for **case 1**. Specifically, to obtain $O(\sqrt{NT})$ regret, DisLinUCB [28] requires $O(d N^{1.5}\log(NT))$ communication,  AsyncLinUCB [17] and FedLinUCB [A1] require $O(d N^{2}\log(NT))$ communication, with the latter two having the same scaling in $N$ as our FedGLB-UCB.

However, with that being said, we can still make some conjecture about the optimal communication cost in order to attain $O(\sqrt{NT})$ regret for the FedGLB problem. As mentioned in line 201 of our paper, for a dataset of size $NT$, e.g., the global dataset over all $N$ clients at time $T$, any distributed optimization method needs at least $\Omega(\sqrt{NT \log{\frac{1}{\epsilon}}})$ rounds of communication to converge to the optimum up to an $\epsilon$ error [3], with each round involving all $N$ clients. If we can prove that in order to obtain optimal regret over time horizon $T$ it is necessary for the FedGLB algorithm to obtain an estimator that is within $\epsilon$ sub-optimality on this global dataset, then this gives us an $\Omega\left( T^{0.5}N^{1.5} \log{\frac{1}{\epsilon}} \right)$ communication lower bound, which indicates there is only a $O(d\sqrt{N})$ gap compared with our proposed FedGLB-UCB.

[A1] He, J., Wang, T., Min, Y. and Gu, Q., 2022. A Simple and Provably Efficient Algorithm for Asynchronous Federated Contextual Linear Bandits. arXiv preprint arXiv:2207.03106.

---

### Author Response · Authors · 2022-08-09
**Common Response to All Reviewers**

We have responded to all questions from our respected reviewers, and highlighted our solution’s technical novelty in Section 4.2 in the revised version, especially how our proof differs from [2,12].

We hope the reviewers find our responses useful and clear out your initial concerns. Please let us know if there is any further information that we can provide to help you better evaluate our work.

---

### Meta-Review · Area_Chair_DtZz · 2022-08-27

**Recommendation:** Accept
**Confidence:** Less certain

**Metareview:**

Federated bandits are a current area of interest within the community and the paper provides valuable contributions. In particular, the authors deal with the rather general GLM setting, provide algorithms, and study the regret. It would be useful if the authors would use the discussions with the reviewers and the reviewers' comments to improve and polish the paper.

**Award:**

No

---

### Decision · Program_Chairs · 2022-09-14

Accept